# Deep Learning-Based Pixel-Wise Lesion Segmentation on Oral Squamous Cell Carcinoma Images

**Francesco Martino** [1] , **Domenico D. Bloisi** [2,*] , **Andrea Pennisi** [3] , **Mulham Fawakherji** [4],
**Gennaro Ilardi** [1], **Daniela Russo** [1], **Daniele Nardi** [4], **Stefania Staibano** [1,†]
**and Francesco Merolla** [5,†]

1    Department of Advanced Biomedical Sciences, University of Naples Federico II, 80131 Napoli, Italy;
    francesco.martino@unina.it (F.M.); gennaro.ilardi@unina.it (G.I.); daniela.russo@unina.it (D.R.);
    stefania.staibano@unina.it (S.S.)
2    Department of Mathematics, Computer Science, and Economics, University of Basilicata,
    85100 Potenza, Italy
3    Allianz Benelux, 1000 Brussels, Belgium; andrea.pennisi@allianz.be
4    Department of Computer Science, Control, and Management Engineering, Sapienza University of Rome,
    00185 Rome, Italy; fawakherji@diag.uniroma1.it (M.F.); nardi@diag.uniroma1.it (D.N.)
5    Department of Medicine and Health Sciences "V. Tiberio", University of Molise, 86100 Campobasso, Italy;
    francesco.merolla@unimol.it
*    Correspondence: domenico.bloisi@unibas.it
†    Co-senior authors.

**Abstract:** Oral squamous cell carcinoma is the most common oral cancer. In this paper, we present a performance analysis of four different deep learning-based pixel-wise methods for lesion segmentation on oral carcinoma images. Two diverse image datasets, one for training and another one for testing, are used to generate and evaluate the models used for segmenting the images, thus allowing to assess the generalization capability of the considered deep network architectures. An important contribution of this work is the creation of the Oral Cancer Annotated (ORCA) dataset, containing ground-truth data derived from the well-known Cancer Genome Atlas (TCGA) dataset.

**Keywords:** oral carcinoma; medical image segmentation; deep learning

## 1. Introduction

Malignant tumors of the head and neck region include a large variety of lesions, the great majority of which are squamous cell carcinomas of the oral cavity [1]. According to GLOBOCAN 2018 data on cancer [2], oral cavity malignant neoplasms, together with lip and pharynx malignancies, account for more than half-million new occurrences per year worldwide, with an estimated incidence of 5.06 cases per 100,000 inhabitants. Moreover, Oral Squamous Cell Carcinoma (OSCC) is characterized by high morbidity and mortality, and, in most countries, the survival rate after five years from the diagnosis is less than 50% of the patients [3].

The histology examination is the gold standard for the definition of these tumors. Surgical pathologists use both clinical and radiological evidence to complement their diagnoses, differentiating between benign and malignant lesions. In the last years, surgical pathology is witnessing a digital transformation thanks to (1) the increase of the processing speed of Whole Slide Images (WSI) scanners [4] and (2) the lower storage costs and better compression algorithm [5]. Consequently, WSI digital analysis is one of the most prominent and innovative topics in anatomical pathology, catching academic and industries attentions. An example of an image obtained using a WSI scanner is shown in Figure 1.

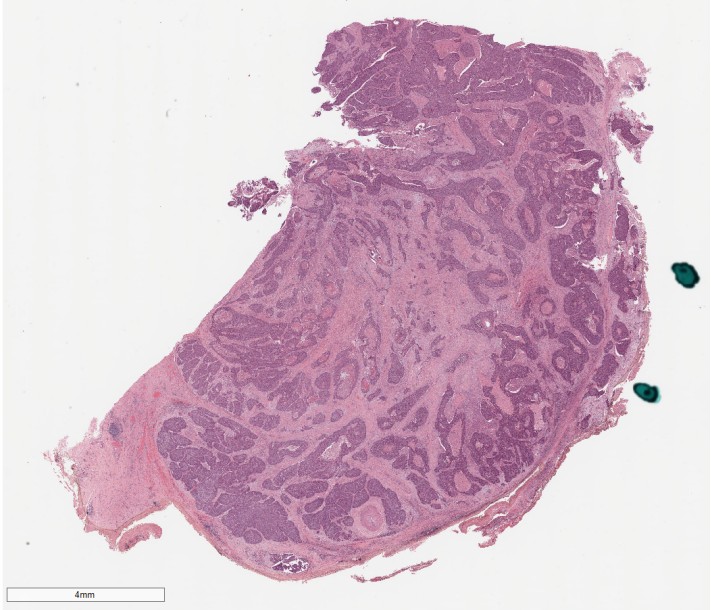

**Figure 1.** An example of image generated by a Whole Slide Images (WSI) scanner. The image has a dimension of 35,862 × 32,195 pixels and the file size is 213.3 MB.

However, WSI (and associated datasets) are characterized by three important limitations:

1. WSI are extremely large images, having a memory size of two gigabytes on average [6].
2. There are a few surgical pathology units that are fully digitalized and that can store a large amount of digitalized slides, although their number is increasing exponentially [7].
3. There is a small number of available image datasets and most of them are not annotated [8].

Due to the above-discussed limitations, the research activity based on Artificial Intelligence (AI) algorithms applied to WSI is still limited compared to other diagnostic imaging branches, such as radiology, but the scientific literature on the topic is growing fast and we are observing the appearance of public datasets of unannotated and annotated histopathology WSI.

In this paper, we present a performance evaluation of four different image segmentation architectures based on deep learning to obtain a pixel-wise separation between benign and malignant areas on WSI samples. In particular, we test four widely used Semantic Segmentation deep neural Networks (SSNs) on publicly available data for the detection of carcinomas. As a difference with respect to classification neural network, SSNs take as input images of arbitrary sizes and produce a correspondingly sized segmented output, without relying on local patches.

The contributions of this work are three-fold:

1. We compare four different supervised pixel-wise segmentation methods for detecting carcinoma areas in WSI using quantitative metrics. Different input formats, including separating the color channels in the RGB and Hue, Saturation, and Value (HSV) models, are taken into account in the experiments.
2. We use two different image datasets, one for training and another one for testing. This allows us to understand the real generalization capabilities of the considered SSNs.
3. We created a publicly available dataset, called Oral Cancer Annotated (ORCA) dataset, containing annotated data from the Cancer Genome Atlas (TCGA) dataset, which can be used by other researchers for testing their approaches.

The paper is organized as follows. Section 2 contains a discussion of similar approaches present in the literature. Section 3 describes the details of the proposed method, while Section 4 shows both qualitative and quantitative results obtained on publicly available data. Finally, conclusions are drawn in Section 5.

## 2. Related Work

Artificial intelligence (AI) algorithms have been proposed to address a wide variety of questions in medicine; e.g., for prostate Gleason score classification [9], renal cancer grading [10], breast cancer molecular subtyping [11] and their outcome prediction. Moreover, AI-based methods have been applied to the segmentation of various pathological lesions in the fields of neuropathology [12], breast cancer [13], hematopathology [14], and nephrology [15].

The above-cited studies have been conducted mainly on the most common tumors (i.e., breast or prostate), while AI-based methods have been scarcely adopted to deal with other types of cancer, despite their high incidence and mortality rates, as the Oral Squamous Cell Carcinoma (OSCC). The analysis of a recent systematic review by Mahmood et al. [16] shows that still few applications of automatic WSI analysis algorithms are available for OSCC. In particular, the survey reports 11 records about the employment of AI-based methods for the analysis of specific histological features of oral lesions: out of 11, only four papers refer to OSCCs, namely [17–20], one paper is about oral epithelial dysplasia [21], five about oral submucous fibrosis, i.e., [22–26], and one paper is about oropharyngeal squamous cell carcinoma [19]. Another recent application of machine learning algorithms on oral lesions histopathological is based on immunohistochemistry (IHC) positivity prediction [27].

Segmentation methods on WSI images have been developed mainly for nuclei segmentation [28–30], epithelium segmentation [19,24], microvessels and nerves [31], and colour-based tumour segmentation [17]. Recently, Shaban et al. [32] proposed an indirect segmentation method, through small tiles classification, with an accuracy of 95.12% (sensitivity 88.69%, specificity 97.37%).

To the best of our knowledge, there are no published results on direct OSCC segmentation using deep learning and none employing the TCGA as a source of histopathological images. This work represents a first attempt of applying well-known deep learning-based segmentation methods on the publicly available TCGA images, providing also annotations to quantitatively validate the proposed approach.

*Datasets*

Concerning WSI datasets, most of them have been made available for challenges, such as Camelyon [33] and HEROHE, on Kaggle or as part of larger databases. The Cancer Genome Atlas (TCGA) [34] contains publicly available data provided by the National Cancer Institute (NCI), which is the U.S. federal government's principal agency for cancer research and training, since 2006. In particular, it contains clinicopathological information and unannotated WSI of over 20,000 primary cancer covering 33 different cancer types [35].

## 3. Methods

Our aim is to use a supervised method to automatically segment an input WSI sample into three classes:

1. Carcinoma pixels;
2. Tissue pixels not belonging to a carcinoma;
3. Non-tissue pixels.

Figure 2 shows the functional architecture of our approach. We worked on input images having a large dimension of $4500 \times 4500$ pixels, which is an input dimension about ten times greater than the input dimension supported by existing segmentation SSNs. Thus, a preprocessing step is needed in order to fit the input format of the deep neural network. We tested two different pre-processing functions:

- Simple resizing, where the original input WSI sample is resized from $4500 \times 4500$ to $512 \times 512$ pixels without any other change in the color model.
- Color model change, where the WSI sample is resized to $512 \times 512$ pixels and the original color model is modified. For example, we tested as input for the deep neural network the use of the Red channel of the RGB model in combination with the Hue channel of the HSV model.

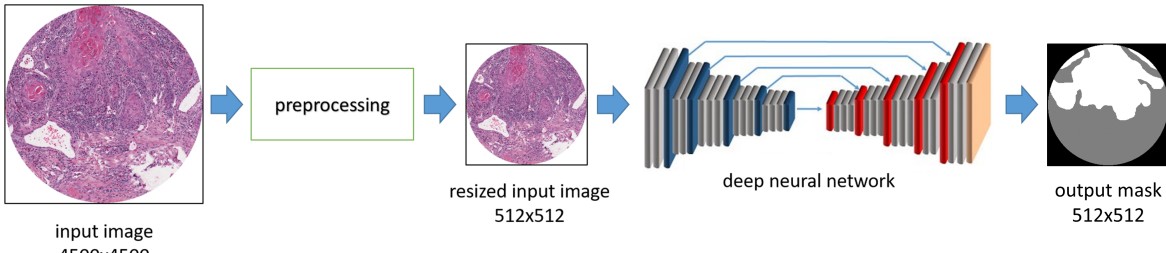

**Figure 2.** Functional architecture of the proposed approach.

Segmentation results obtained using the different pre-processing functions are discussed in Section 4.

### 3.1. Network Architectures

The core of the proposed approach consists in the use of a semantic segmentation network. We have selected four different network architectures among the most used ones:

1. SegNet [36].
2. U-Net [37].
3. U-Net with VGG16 encoder.
4. U-Net with ResNet50 encoder.

#### 3.1.1. Segnet

SegNet is made of an encoder network and a corresponding decoder network, followed by a final pixel-wise classification layer (Figure 3). The encoder network consists of the first 13 convolutional layers of the VGG16 [38] network designed for object classification, without considering the fully connected layer in order to retain higher resolution feature maps at the deepest encoder output. In such a way, the number of parameters to train is significantly reduced. Each encoder layer has a corresponding decoder made of 13 layers. The final decoder output is fed to a multi-class soft-max classifier to produce class probabilities for each pixel independently.

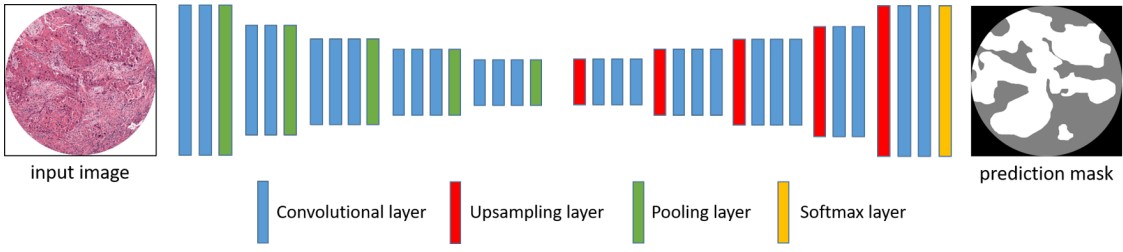

**Figure 3.** SegNet architecture.

#### 3.1.2. U-Net

The architecture of the net is shown in Figure 4. The input image is downsampled to obtain a $512 \times 512$ resized image.

The encoding stage is needed to create a 512 feature vector and it is made of ten $3 \times 3$ convolutional layers, and by four $2 \times 2$ max pooling operations with stride 2. In particular, there is a repeated application of two unpadded convolutions, each followed by a rectified linear unit (ReLU) and a max pooling operation. The decoding stage (see the right side of Figure 4) is needed to obtain the predicted mask at $512 \times 512$ pixels. It is made of eight $3 \times 3$ convolutional layers and by four $2 \times 2$ transpose layers. There is a repeated application of two unpadded convolutions, each followed by a ReLU and a transpose operation. Figure 4 shows also the concatenation arcs from the encoding side to the decoding side of the network. Cropping is necessary due to the loss of border pixels in every convolution layer.

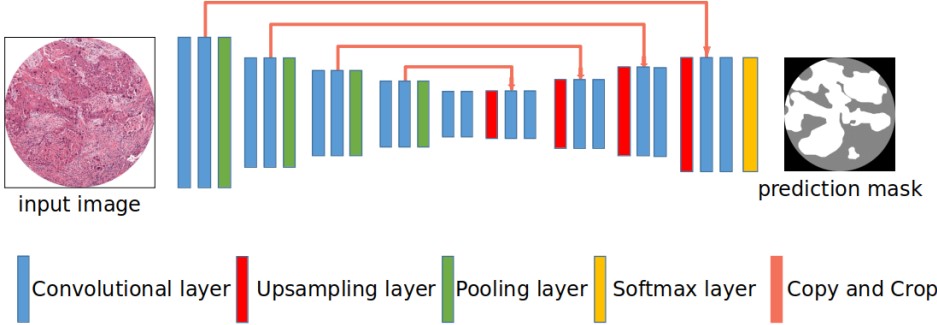

**Figure 4.** U-Net architecture.

### 3.1.3. U-Net with Different Encoders

The original U-Net consists of two paths, the first one (left side) composed of four blocks. Each block consists of a typical architecture of a convolution network, composed of repeated two $3 \times 3$ Convolutional layers followed by a rectified linear unit (ReLU) and a $2 \times 2$ max pooling layer with stride 2 as down-sampling steps. The second path (right side) represents the up-sampling path also composed of four blocks. Each block consists of an up-sampling feature map followed by concatenation with the corresponding cropped feature map from the first path followed by a double $3 \times 3$ Convolutional layer. Each convolution layer is composed of batch normalization. Between the first and the second path, there is a bottleneck block. This bottleneck is built from two convolution layers with batch normalization and dropout. In the models we applied, we replaced the first path, which is the encoder path of the network, with two different models, namely VGG16 [38] and ResNet50 [39], in order to obtain higher accuracy.

### 3.2. Training and Test

Two diverse datasets are used to train, validate, and test the networks. The training dataset consists of 188 annotated images of advanced OSCC derived from the digital data acquired in the Surgical Pathology Unit of the Federico II Hospital in Naples (Italy). The study was performed in agreement with Italian law. Moreover, according to the Declaration of Helsinki for studies based only on retrospective analyses on routine archival FFPE-tissue, written informed consent was acquired from the living patient at the time of surgery. The validation dataset consists of 100 annotated images from the newly created ORCA dataset (described below), while the test dataset consists of a further 100 images from the ORCA dataset also. All the images in the two datasets have been manually annotated by two of the authors of this paper that are expert pathologists (D. Russo and F. Merolla, both MD PhD and Board in Pathology) using three color labels:

1. Carcinoma pixels, colored in white.
2. Tissue pixels not belonging to a carcinoma, colored in grey.
3. Non-tissue pixels, colored in black.

The above-listed labels represent the classes learned by the deep network during the training stage. Figure 5 shows an example of annotation mask.

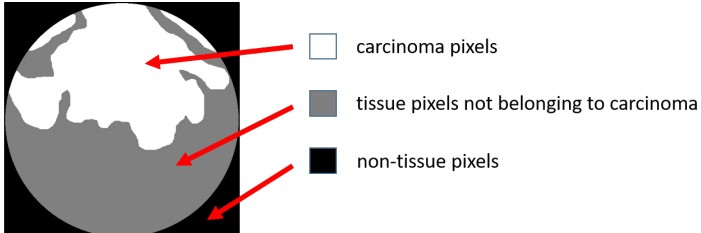

**Figure 5.** Example of annotation mask. Carcinoma pixels are colored in white, tissue pixels not belonging to a carcinoma are colored in grey, and non-tissue pixels are colored in black.

### 3.2.1. Training Data

The dataset used for training consists of a set of images collected by the Surgical Pathology Unit of the Federico II Hospital, in Naples (Italy). In particular, cases were assembled as four Tissue Micro Array and, after H&E staining, the slides were scanned with a Leica Aperio AT2 at 40× (20× optical magnification plus a 2× optical multiplier).

Slides were annotated using the Leica Aperio ImageScope software. In the first instance, to define each core, we used the ImageScope rectangle tool with a fixed size of 4500 × 4500 pixels, corresponding to nearly 1.125 mm, i.e., the size of each TMA core. Then, an expert pathologist used the pen tool to contour the tumor areas within each core. ImageScope provides an XML file as output for each WSI. RGB core images were extracted using OpenCV and OpenSlide libraries.

Concerning the annotation masks, we started by automatically detecting the tissue portion of each core using the OpenCV functions contouring and convex hull, to isolate the tissue pixels (colored in grey, as mentioned beforehand) from the background (non-tissue pixels colored in black). Then, the tumor masks (colored in white) were superimposed over the tissue pixels to draw carcinoma pixels. Finally, the masks were manually refined at pixel-size resolution.

An example of a couple <original image, annotated mask> used for the training stage is shown on the top of Figure 6.

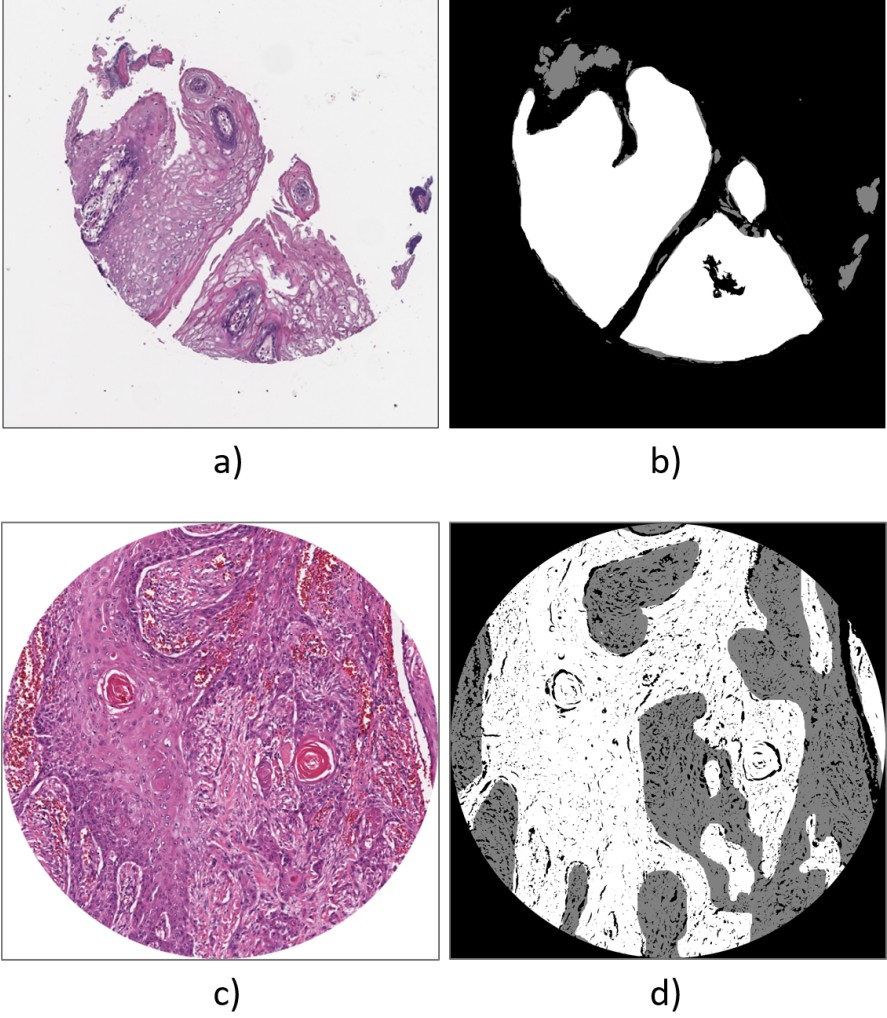

a)　　　　　　　　　　　　　　　　b)

c)　　　　　　　　　　　　　　　　d)

**Figure 6.** Top: Example of annotated data from the training dataset. (**a**) Original image. (**b**) Manually generated annotation mask. Bottom: Example of annotated data from the Oral Cancer Annotated (ORCA) dataset. (**c**) Original image. (**d**) Manually generated annotation mask.

### 3.2.2. ORCA Dataset

An important contribution of this work is the creation of a fully annotated dataset, called ORCA, consisting of couples of WSI samples plus corresponding annotation masks. An example of a couple <original image, annotated mask> from the ORCA dataset is shown on the bottom of Figure 6.

The original WSI samples belong to the Cancer Genome Atlas (TCGA), a landmark cancer genomics program, molecularly characterized over 20,000 primary cancer and matched normal samples spanning 33 cancer types [40]. TCGA data are publicly available for anyone in the research community to use. For each WSI included in the TCGA dataset, we defined one or two cores containing a representative tumor area. The dataset contains WSI scanned both at 20× and at 40×.

TCGA whole slide images were annotated using the Leica Aperio ImageScope software. For each image, a circular core region has been selected using the ImageScope ellipse tool, with a fixed size of 4500 × 4500 pixels in the case of the WSI scanned at 40× or 2250 × 2250 pixels in the case of WSI acquired at 20×. This allowed us to keep the same spatial dimension for all the selected cores. Then, the tumor contours have been drawn using a pen tool. The RGB images and the corresponding annotation masks were extracted using OpenCV and OpenSlide libraries.

All the couples of WSI samples plus corresponding annotation masks used in this work are available for downloading from the ORCA dataset web page at: https://sites.google.com/unibas.it/orca.

## 4. Experimental Results

In order to train and evaluate the networks, we split the image data into training, validation, and test sets. The training set consists of the 188 images from the Federico II Hospital plus a set of images obtained by applying a simple augmentation technique. This allows for augmenting the number of the available training samples. In particular, the data augmentation has been achieved by flipping the images vertically, horizontally, and in both ways. The final cardinality of the augmented training set is 756 images.

Each validation and test set includes 100 images from the ORCA data set. It is worth noticing that validation and test set are completely disjoint sets. In such a way, we tested the capability of the networks in generalizing the segmentation problem.

As stated above, we trained four different models:

- SegNet.
- U-Net.
- U-Net with VGG16 encoder.
- U-Net with ResNet50 encoder.

The source code of the used SegNet network is available at https://github.com/apennisi/segnet, while the source code for U-Net and its modifications is available at https://github.com/Mulham91/Deep-Learning-based-Pixel-wise-Lesion-Segmentationon-Oral-Squamous-Cell-Carcinoma-Images.

The four networks have been trained by using the Adam optimizer, with a learning rate of $1e - 4$, without adopting any decay technique. The input image size has been set to 512 × 512 pixels.

For computing the loss, we used the Cross-Entropy function [41], which is a loss function widely used in deep learning. Cross entropy helps in speeding up the training for neural networks in comparison to the other losses [42]. The cross entropy for multi-class error computation is defined as follows:

$$s = -\sum_{c=1}^{M} y_{o,c} \log(p_{o,c}), \tag{1}$$

where $c$ is the class id, $o$ is the observation id, and $p$ is the probability. Such a definition may also be called categorical cross entropy.

The training has been manually stopped after 60 epochs for all the considered network architectures because we noticed a trend of all the network in overfitting the training set.

We performed three kinds of experiments:

1.  Using RGB images as input;
2.  Taking into account the HSV color representation and concatenating the Hue channel with the Red channel from the RGB space;
3.  Using the Red and Value channels.

An example of the different color models mentioned above is shown in Figure 7. The original RGB image is decomposed into single channels using the RGB and HSV color models. Then, some channels have been selected as input for the deep network. In particular, we used H + R and R + V.

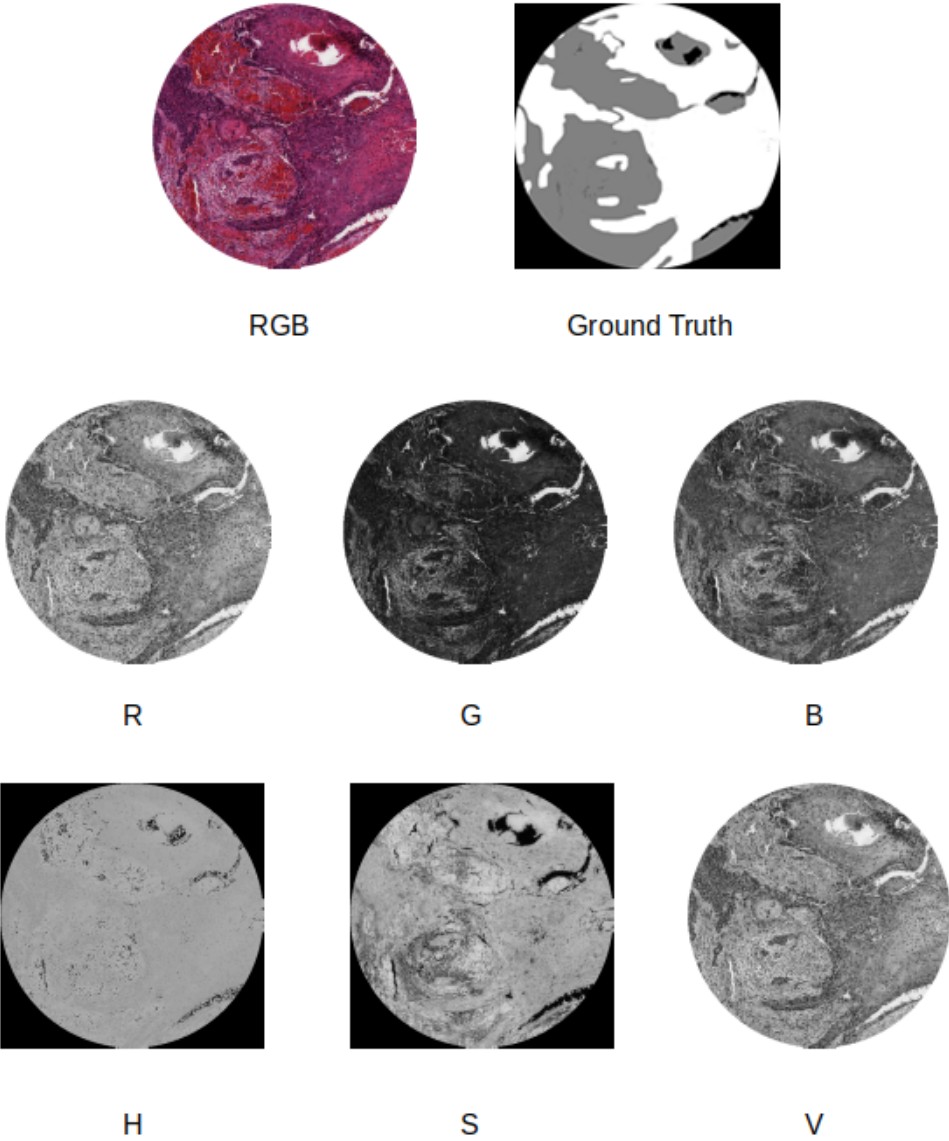

**Figure 7.** An example of the input images used for training the networks. First row: Original input and corresponding annotated mask. Second row: Red, Green, and Blue channels from the original RGB image. Third row: Hue, Saturation, and Value channels from the transformation of the original RGB image into Hue, Saturation, and Value (HSV).

The idea of using a multi-spectral input derives from the application of deep learning techniques in the area of precision farming, where multi-spectral inputs have denoted good performance on SSNs for the segmentation of crop and weed plants in images acquired from farming robots [43].

The prediction masks produced by the deep networks have been compared with the ground-truth annotation masks in order to measure the capability of our pipeline in generating accurate results. In particular, we have used the number of false-positive (FP), false-negative (FN), and true-positive (TP) carcinoma pixel detections as an indicator for the evaluation of the results. Figure 8 shows an example of how we computed FP, FN, and TP pixels to evaluate the predicted mask.

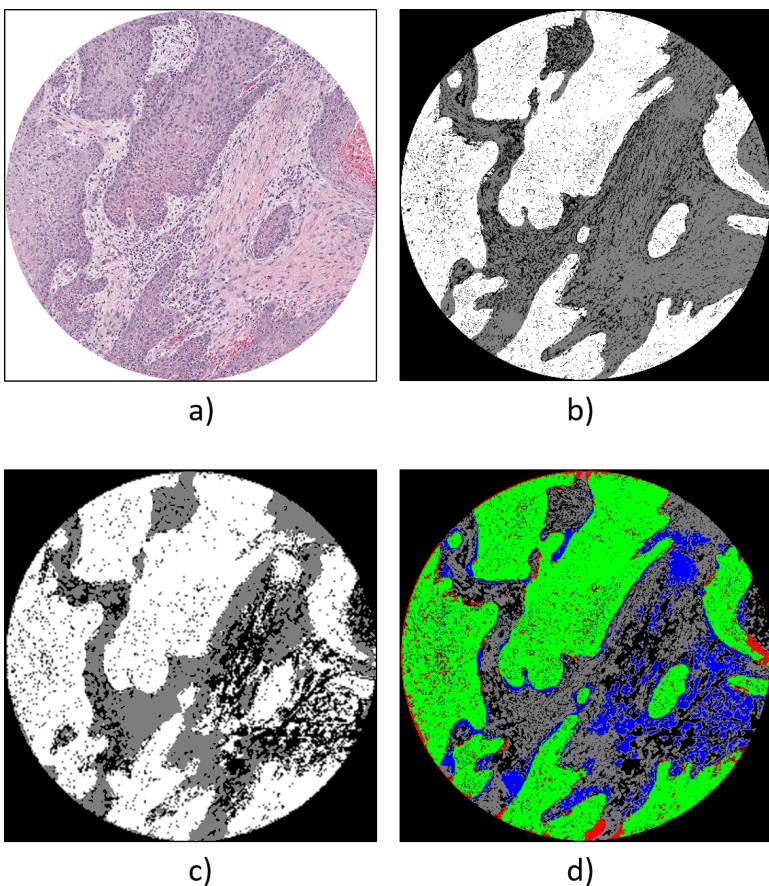

**Figure 8.** An example of error analysis for a test image. (**a**) The original image included in the test set. (**b**) The corresponding annotation mask, where carcinoma pixels are coloured in white. (**c**) The predicted mask generated from U-Net with ResNet50 encoder. (**d**) The coloured error mask, where green pixels are true-positive (TP) carcinoma pixels, blue are false-positives (FPs), and red are false-negatives (FNs).

*4.1. Qualitative Evaluation*

We evaluated the output of the trained network, in the first instance, by visual inspection. This allowed us to interpret the results and judge them in the light of the pathologist's experience and diagnostic reasoning in histopathology.

In most cases, we observed that the areas reported as false positives (blue pixels in Figure 8d) actually corresponded to small nests of stromal infiltration, located in correspondence with the tumor invasion front.

Vice versa, we observed that a portion of the pixels reported as false negatives actually referred to small areas of stroma within the tumor area.For example, by observing the colored mask in Figure 8d, it is visible that the stromal area within the tumor is considered as a set of false-negative pixels (coloured in red) even if they should be considered as true positives.

Given the discussion above, from a qualitative point of view, the algorithm has often reported pixels relating to areas of intense inflammatory infiltrate as false positives. However, single interspersed lymphocytes in the peritumor stroma were not per se a problem in the interpretation of the image.

Tumor grading was another determining factor in the efficiency of the algorithm. In fact, the trained network recognized with difficulty the highly differentiated tumor areas, with a prevalence of keratin pearls. This last factor could be attributable to the fact that the dataset used for the training was mainly composed of a series of high grade advanced squamous carcinomas.

### 4.2. Quantitative Results

To obtain a quantitative measure of the segmentation performance achieved by the four deep networks, we used the Mean Intersection-Over-Union (mIOU) metric. MIOU is one of the most common metrics for evaluating semantic image segmentation tasks. In particular, we first computed the IOU for each semantic class, namely carcinoma, tissue non-carcinoma, and non-tissue, and then computed the average over all classes. The *mIOU* is defined as follows:

$$mIOU = \frac{1}{C} \sum_{i=1}^{C} \frac{TP_j}{TP_j + FP_j + FN_j}, \tag{2}$$

where *TP* is true-positive, *FP* is false-positive, *FN* is false-negative, and *C* is the total number of classes.

Table 1 shows the quantitative results of the semantic segmentation on three different inputs: RGB, (Red + Hue), and (Red + Value). The results show that the use of the combination (Red + Value) generates better results than (Red + Hue) input. Moreover, a deeper network, such as U–Net modified with ResNet50 as encoder, performs better than the original U–Net (having a more shallow encoder).

**Table 1.** Pixel-wise segmentation results.

| Training | SSN | MIoU | IOU | | |
| Input | Type | | Non-Tissue | Tissue Non-Carcinoma | Carcinoma |
|---|---|---|---|---|---|
| RGB | SegNet | 0.51 | 0.74 | 0.49 | 0.30 |
| | U-Net | 0.58 | 0.79 | 0.52 | 0.45 |
| | U-Net + VGG16 | 0.64 | 0.84 | 0.56 | 0.45 |
| | U-Net + ResNet50 | 0.67 | 0.85 | 0.59 | 0.56 |
| Red + Hue | SegNet | 0.49 | 0.70 | 0.48 | 0.30 |
| | U-Net | 0.55 | 0.78 | 0.50 | 0.38 |
| | U-Net + VGG16 | 0.33 | 0.22 | 0.28 | 0.53 |
| | U-Net + ResNet50 | 0.57 | 0.80 | 0.48 | 0.43 |
| Red + Value | SegNet | 0.54 | 0.72 | 0.49 | 0.40 |
| | U-Net | 0.57 | 0.78 | 0.49 | 0.46 |
| | U-Net + VGG16 | 0.62 | 0.79 | 0.50 | 0.55 |
| | U-Net + ResNet50 | 0.63 | 0.80 | 0.52 | 0.56 |

### 4.3. Discussion

The histological evaluation of Hematoxylin Eosin stained slides from tumor samples, carried out by an experienced pathologist on an optical microscope, is a mandatory step in the diagnostic, prognostic and therapeutic pathway of patients suffering from squamous cell carcinoma of the oral cavity. To date, for the histological diagnosis of OSCCs, the gold standard is the visual analysis of histological preparations, stained with hematoxylin and eosin; tumor recognition basically takes place on the basis of the qualitative assessment of architectural characteristics of the neoplastic tissue, based on the wealth of knowledge pathologist's own. The qualitative assessment is subjective and can suffer from inter-individual variability, especially in borderline situations that are difficult to interpret. The use of a segmentation algorithm could minimize the interpretative variability and speed up the pathologists' work, providing they with a screening tool, particularly useful in those cases in which the histopathological diagnosis must be carried out on the extensive sampling of complex surgical samples that involve the generation of multiple blocks of formalin-fixed and paraffin-embedded tissue samples, from which numerous slides stained with hematoxylin and eosin are obtained.

Based on the presented results, our contribution is composed of: (i) a novel dataset, the ORCA set, which will allow us to conduct new studies on Oral Squamous Cell Carcinoma. Particularly, the dataset is composed of annotation from the TCGA dataset, a full comprehensive dataset enriched, as well as with diagnostic slide, with clinicopathological information and molecular biology data. This could facilitate the development of molecular characterization deep learning algorithms; (ii) our method relies on $2250 \times 2250$ and $4500 \times 4500$ images, without a tiling processing. Even though an improvement in its accuracy is mandatory for clinical practice, the utilization of so large images can hugely reduce time-demand for a WSI, making our approach easily scalable to clinical routine, when hundreds of slides need to be processed each day; (iii) after demanded improvements and a clinical trial, this kind of algorithm may be part of clinical practice via L.I.S. integration, fastening OSCC diagnosis and helping pathologists to identify OSCC areas on WSI. Indeed, we foresee to extend our method on lymphonodal metastasis, giving the pathologist an easy way to detect small tumor islands, and on distant metastasis, supporting the pathologist with cases of suspect metastasis of OSCC primary tumor.

We intend to propose this artificial intelligence algorithm as a Computer-Aided Diagnostic, aware that it cannot replace the pathologist in his routine activity, but that it will be able to provide they with valid help, especially for those who find themselves working in generalist diagnostic centres on the territory, not specialized in the diagnosis of an infrequent but extremely lethal disease.

## 5. Conclusions

In this work, we created a dataset called ORCA, containing annotated data from the TCGA dataset, to compare four different deep learning-based architectures for oral cancer segmentation, namely: SegNet, U-Net, U-Net with VGG16 encoder, and U-Net with ResNet50 encoder. The peculiarity of this work consists of the use of a training set completely different from the test data. In such a way, we tested the capability of the networks in generalizing the problem, providing promising segmentation results.

Despite the non-optimal results, to the best of our knowledge, this is the first attempt to use an automatic segmentation algorithm for oral squamous cell carcinoma and it represents an important novelty to this pathology. Furthermore, the publically-available ORCA dataset will facilitate the development of new algorithms and will boost the research on computational approaches to OSCC.

As future directions, we will aim at enlarging the training set and at making it publicly available. In this work, we considered color transformation by using a combination of HSV and RGB color models as a method for creating a multi-channel input. This was done because the group of pathologists that are authors of this work noticed that HSV color space contains a lot of visually distinguishing features about tumor cells. We did not use color modifications for augmenting the data. However, this is an interesting aspect that will be investigated in future work. Moreover, we foresee to improve our model to achieve a result that may be transferred to clinical practice.

**Author Contributions:** D.D.B., A.P. and F.M. (Francesco Merolla) conceived and designed the experiments; M.F. and F.M. (Francesco Martino) performed the experiments; F.M. (Francesco Merolla) and D.R. analyzed the data; G.I. contributed reagents/materials/analysis tools; S.S. and D.N. provide a critical review of the paper. All authors have read and agreed to the published version of the manuscript.

**Funding:** Our research has been supported by a POR Campania FESR 2014-2020 grant; "Technological Platform: eMORFORAD-Campania" grant PG/2017/0623667.

**Acknowledgments:** We thank Valerio Pellegrini for his contribution in the annotation of the dataset images.

**Conflicts of Interest:** The authors declare no conflict of interest.

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
