# Peer review of "Deep Learning-Based Pixel-Wise Lesion Segmentation on Oral Squamous Cell Carcinoma Images"

_applsci, doi:10.3390/app10228285_

Round 1

Reviewer 1 Report

It should be considered changing the title of work: automatic methods are something different than machine learning, by the concept of automatic methods we mean in this case classic image processing methods, most often with automatically matched parameters.

Disadvantages

1. This work is too much focused on Oral squamous cell carcinoma. Machine learning has been successful with varying degrees of success for many tissues, notably the multi-organ / tissue approach is a challenge.

2. I'm afraid that the manual mask used here is too general and inaccurate to use the pixel-by-pixel quality assessment method for segmentation,

3. Before the black box was taught, was it checked how color operations such as color normalization, color transfer or deconvolution methods do not distinguish the tissue you are looking for? Because at a glance you can see where the changes are occurring. Maybe an iso-map of features would be a good differentiator?

4. The State-of-the-Art can be extended.

Advantages

The advantages of the article should also be indicated. A rather interesting approach, which has a positive effect on the results, is to use channels from different color spaces. Most of the methods focus only on RGB, there are also some that only work with HSV, while the classic approach is color deconvolution (e.g. Macenko, Khan).

Reviewer 2 Report

In this article, the authors are providing an annotated histopathology dataset from TCGA for oral squamous cell carcinoma. They used machine learning algorithms to classify pathological images. With the abundance of digital histopathological images, this article's results could contribute to a better understanding of the role of artificial intelligence in the analysis of this type of data. Overall the paper is written clearly. Followings are my specific comments to the authors:

  1. Could the authors clarify if the labeling of the annotated images was performed by a pathologist(s) or not?
  2. It looks like the ground truth annotated images are showing very rough labeling (i.e., the labels are not covering details in the image). In this case, areas that are miss labeled in the image will be wrongly classified later. Why is the annotation not performed in a higher resolution, and how would the authors justify that?
  3. Why the authors used only orientation for data augmentation? Should shifting color (color augmentation) be added for that purpose, too? Why or why not?
  4. The authors are reducing the magnification of the images to improve data storing. How much effect would this have on the final decision making?
  5. In the annotated example by the authors (Figure 6), on the top left, there is an area that seems to be non-tissue but is labeled as cancer tissue, why is this happening, and how often this has happened in the annotations?
  6. Could the authors clarify what IOU implies? Which model is doing better overall?
  7. Metrics like sensitivity, specificity, and Area under the receiver operative characteristic curve might be easier for readers to perceive. Could the authors add that metric to their quantitative evaluations as well?
  8. Could the authors elaborate more on their findings and their specific contribution to the field? How their results may be used by clinicians to analyze pathological images? What are the limitations and flaws of their method? In general, the article is missing a discussion section.

Other comments

  1. Line 59-71, there is no need to include the references in parenthesis: (e.g., []). You can directly bring the reference after it is mentioned.
  2. Figure 5 to figure 7 can be shown in a single figure to reduce the number of figures.

Round 2

Reviewer 1 Report

Responses to the comments are satisfactory.
The shared dataset is a very valuable element of the article and enhances its contribution to research.

Author Response

Manuscript ID: applsci-978655

Original title: Automatic Lesion Segmentation on Oral Squamous Cell Carcinoma Images

New title: Deep Learning-based Pixel-wise Lesion Segmentation on Oral Squamous Cell Carcinoma Images

Authors: Francesco  Martino,  Domenico  Bloisi*,  Andrea  Pennisi,  Mulham Fawakherji,  Gennaro Ilardi,  Daniela Russo,  Daniele Nardi,  Stefania Staibano, Francesco Merolla

Journal: Applied Sciences

*Corresponding author:domenico.bloisi@unibas.it

Second revision

The authors wish to thank the reviewers for their effort in reviewing the revised version of the manuscript. The paper has been modified in accordance to the reviewers’ suggestions about the revised version of the text. Kindly find our responses to the reviewers’ comments below (reviewers’ comments are indicated in italic font).

Reviewer:

"The encoding stage is needed to create a 512 feature vector, while the decoding stage is needed to obtain the predicted mask at 512×512 pixels." Generally, U-Net has several connections between the encoding stage and the decoding stage. Please add the description and image of the connections to main text and Figure 4, respectively”.

Authors:

Figure 4 has been modified to add the connections between the encoding and decoding stage. Moreover, the following new paragraph has been added on page 5:

“Figure 4 shows also the concatenation arcs from the encoding side to the decoding side of the network. Cropping is necessary due to the loss of border pixels in every convolution layer.”

Reviewer:

"Training data consists of 188 annotated images of advanced OSCC derived from the digital data acquired in the Surgical Pathology Unit of the Federico II Hospital, in Naples (Italy)." I speculate that this dataset is not public. If so, please describe IRB approval and informed consent in the paper.”

Authors:

The study was performed in agreement with the Italian law, and according to the Declaration of Helsinki, for studies based only on retrospective analyses on routine archival FFPE-tissue; a written informed consent was acquired from the living patient at the time of surgery. A paragraph about that has been added on page 5.

Reviewer:

"All the images in the two datasets have been manually annotated by expert pathologists" Please clarify the initials and affiliations of expert pathologists and their experiences as pathologists.”

Authors:

All the images in the two datasets have been manually annotated by F. Merolla and D. Russo, both MD PhD and Board in Pathology. A new paragraph about that has been added on page 5.

Here a list of publications of our pathologists in the field of oral cancer pathology:

1: Martino F, Varricchio S,  Russo D,  Merolla F, Ilardi G, Mascolo M, dell'Aversana GO, Califano L, Toscano G, Pietro G, Frucci M, Brancati N, Fraggetta F, Staibano S. A Machine-learning Approach for the Assessment of the Proliferative Compartment of Solid Tumors on Hematoxylin-Eosin-Stained Sections. Cancers (Basel). 2020 May 25;12(5):1344. doi: 10.3390/cancers12051344. PMID: 32466184; PMCID: PMC7281627. 

2: Morra F,  Merolla F, Picardi I,  Russo D, Ilardi G, Varricchio S, Liotti F, Pacelli R, Palazzo L, Mascolo M, Celetti A, Staibano S. CAF-1 Subunits Levels Suggest Combined Treatments with PARP-Inhibitors and Ionizing Radiation in Advanced HNSCC. Cancers (Basel). 2019 Oct 17;11(10):1582. doi: 10.3390/cancers11101582. PMID: 31627329; PMCID: PMC6827109. 

3:  Russo D,  Merolla F, Varricchio S, Salzano G, Zarrilli G, Mascolo M, Strazzullo V, Di Crescenzo RM, Celetti A, Ilardi G. Epigenetics of oral and oropharyngeal cancers. Biomed Rep. 2018 Oct;9(4):275-283. doi: 10.3892/br.2018.1136. Epub 2018 Jul 27. Erratum in: Biomed Rep. 2020 May;12(5):290. PMID: 30233779; PMCID: PMC6142034. 

4: Ilardi G,  Russo D, Varricchio S, Salzano G, Dell'Aversana Orabona G, Napolitano V, Di Crescenzo RM, Borzillo A, Martino F,  Merolla F, Mascolo M, Staibano S. HPV Virus Transcriptional Status Assessment in a Case of Sinonasal Carcinoma. Int J Mol Sci. 2018 Mar 16;19(3):883. doi: 10.3390/ijms19030883. PMID: 29547549; PMCID: PMC5877744. 

5: Caroppo D, Salerno G,  Merolla F, Mesolella M, Ilardi G, Pagliuca F, De Dominicis G, Califano L, Ciancia G,  Russo D, Mascolo M. Coexistent Squamous Cell Carcinoma and Granular Cell Tumor of Head and Neck Region: Report of Two Very Rare Cases and Review of the Literature. Int J Surg Pathol. 2018 Feb;26(1):47-51. doi: 10.1177/1066896917724513. Epub 2017 Aug 7. PMID: 28783989. 

6:  Russo D,  Merolla F, Mascolo M, Ilardi G, Romano S, Varricchio S, Napolitano V, Celetti A, Postiglione L, Di Lorenzo PP, Califano L, Dell'Aversana GO, Astarita F, Romano MF, Staibano S. FKBP51 Immunohistochemical Expression: A New Prognostic Biomarker for OSCC? Int J Mol Sci. 2017 Feb 18;18(2):443. doi: 10.3390/ijms18020443. PMID: 28218707; PMCID: PMC5343977. 

7:  Merolla F, Mascolo M, Ilardi G, Siano M,  Russo D, Graziano V, Celetti A, Staibano S. Nucleotide Excision Repair and head and neck cancers. Front Biosci (Landmark Ed). 2016 Jan 1;21:55-69. doi: 10.2741/4376. PMID: 26709761. 

8: Pannone G, Bufo P, Pace M, Lepore S,  Russo GM, Rubini C, Franco R, Aquino G, Santoro A, Campisi G, Rodolico V, Bucci E, Ilardi G, Mascolo M,  Merolla F, Lo Muzio L, Natalicchio I, Colella G, Laurenzana I, Trino S, Leonardi R, Bucci P. TLR4 down-regulation identifies high risk HPV infection and integration in head and neck squamous cell carcinomas. Front Biosci (Elite Ed). 2016 Jan 1;8:15-28. PMID: 26709642. 

9: Caroppo D,  Russo D,  Merolla F, Ilardi G, Del Basso de Caro M, Di Lorenzo P, Varricchio S, Mascolo M, Staibano S. A rare case of coexistence of metastasis from head and neck squamous cell carcinoma and tuberculosis within a neck lymph node. Diagn Pathol. 2015 Oct 29;10:197. doi: 10.1186/s13000-015-0430-x. PMID: 26510425; PMCID: PMC4625527. 

10: Mascolo M, Ilardi G,  Merolla F,  Russo D, Vecchione ML, de Rosa G, Staibano S. Tissue microarray-based evaluation of Chromatin Assembly Factor-1 (CAF-1)/p60 as tumour prognostic marker. Int J Mol Sci. 2012;13(9):11044-62. doi: 10.3390/ijms130911044. Epub 2012 Sep 5. PMID: 23109837; PMCID: PMC3472729. 

11: Mascolo M, Siano M, Ilardi G,  Russo D,  Merolla F, De Rosa G, Staibano S. Epigenetic disregulation in oral cancer. Int J Mol Sci. 2012;13(2):2331-53. doi: 10.3390/ijms13022331. Epub 2012 Feb 21. PMID: 22408457; PMCID: PMC3292026.